# Testing of Lubricants for DIC Tests to Measure the Forming Limit Diagrams of Aluminum Thin Sheet Materials

**Szabolcs Szalai** [1], **Hanna Csótár** [1], **Dmytro Kurhan** [2], **Attila Németh** [1,*], **Mykola Sysyn** [3]
**and Szabolcs Fischer** [1,*]

1 Central Campus Győr, Széchenyi István University, H-9026 Győr, Hungary
2 Department of Transport Infrastructure, Ukrainian State University of Science and Technologies, UA-49005 Dnipro, Ukraine
3 Department of Planning and Design of Railway Infrastructure, Technical University Dresden, D-01069 Dresden, Germany
* Correspondence: nemeth.attila@sze.hu (A.N.); fischersz@sze.hu (S.F.); Tel.: +36-(96)-613-544 (S.F.)

**Abstract:** We investigated lubricants and thin teflon foils that can be applied in the formability testing of the thin aluminum sheets used in the electronics and automotive industries. For the tests, thirteen different industrial lubricants (oils and greases) (i.e., L1–L13) and two Teflon films (i.e., 0.08 and 0.22 mm thick) were applied. The authors conducted an Erichsen test, and the thickness reduction of the discs was measured first. In forming-limit curve (FLC) tests, it is crucial that the stresses are localized in the central area of the specimen during forming and that biaxial deformation is maintained throughout, if possible. We aimed to achieve and fulfill this task. To perform this measurement, the GOM ARAMIS measuring system was utilized. It is an optical measuring system based on the digital image correlation (DIC) principle, capable of measuring both stresses and displacements in real time. A specific validation method was also developed to qualify the DIC system. We concluded that there was a 5% difference in Erichsen indentation (*IE*) number diagrams between the best (L12) and worst (L4) cases for the lubricants and oils tested, which is a significant difference for thin plates. We found that this value could be increased and improved by using Teflon discs. Furthermore, the localization of stress maxima, i.e., the centering of cracks in the specimen, could be achieved by combining Teflon discs and L12 lubricant (with appropriate layer order), which significantly aids in the recording of standard FLC diagrams. Using foils is also advantageous because they are readily available, have no expiry date, and are of less environmental concern.

**Keywords:** DIC; GOM ARAMIS; thin sheets; electrical aluminum; lubricants; FLC; Erichsen test

## 1. Introduction

The DIC technique was developed in the 1980s at the University of North Carolina. However, it was not until the 2000s that image correlation began to become popular in parallel with the development of cameras. Today, DIC techniques are used in the vast majority of published studies instead of traditional strain measurement methods. The technology is noncontact, material-independent, and easily adaptable to various research tasks. The present study dealt with the application of DIC technology in recording deformation boundary diagrams. One of the most critical aspects of forming limit diagrams is the evolution of the friction between the sheet material and tool. In standard tests (ISO 12004-2:2008—Metallic materials—Sheet and strip—Determination of forming-limit curves—Part 2: Determination of forming-limit curves in the laboratory) [1], the standard does not specify a precise method for reducing friction: only the location of the crack on the specimen surface and its tolerance are specified. Under optimum friction conditions, the crack appears at the specimen's center or apex. If it deviates from this, then the friction conditions are not appropriate. Considering this, in the present research, we looked for

suitable lubricants or combinations to keep the friction conditions at the optimum value. That is, the aim was to have cracks at the apex of the specimen.

In addition to corrosion protection, lubricants play an important role in reducing friction between contact surfaces with [2] or without nanoparticles [3,4], for example, during metal forming. By reducing friction, tool lifetime can be increased, making sheet metal forming operations more cost-effective. As a result, many studies have been published on the subject, with different approaches to using lubricants and their composition [5,6]. The most popular and promoted topic in the 21st century is integrating environmental awareness into everyday life, which has already been integrated in industrial areas. Owing to this approach, research has been launched to find alternatives to various kinds of lubricating oil that offer excellent lubricating properties and can be used in an environmentally friendly way. In addition to environmentally friendly lubricants, the role of solid friction-improving materials, which can be reused, is dynamically growing in the field of sheet metal forming. Therefore, an essential aspect of our literature review was examining alternative lubricants to those traditionally used in industrial practice [6].

In Yoshimura et al. [7], wheat flour was suspended in water at different concentrations to use these suspensions as lubricants on titanium and stainless steel as well as mild steel plates. The performance of the lubricants was evaluated using the Erichsen cupping test, measured by scanning electron microscopy. In this way, an environmentally friendly method of friction reduction was achieved, with excellent lubrication performance [7].

In Rao and Wei [8], the usability and lubricity of boric acid dry films in cold-forming operations of aluminum alloys were investigated. During drawing and stretching operations, the results showed that boric acid films are comparable to commonly used solid and liquid lubricants in terms of their lubricity over a wide range of forming speeds. Given the results, it was found that boric acid film firmly adhered to the surface after applying an ethanol solution. Furthermore, it was observed in the scanning electron microscopic (SEM) images that no forming microscratches were visible in the case of boric acid lubrication after technological operations. Finally, as boric acid is nontoxic and insoluble in water, the post-cleaning of formed parts was made relatively easy and safe by using this alternative [8].

Later, Rao et al. [9] continued research on the effectiveness of boric acid as a lubricant in ring-forging and deep-drawing operations. The studies revealed that the process waiting time between the application of the boric acid coating and the forming process significantly affects the friction coefficient. So, fresh surfaces can provide the least friction. However, the long waiting time allows moisture to be absorbed from the ambient air, thus increasing friction. In contrast, the forming speed does not affect the performance of boric acid lubricant, and tests of tools with different friction-reducing coatings have shown that the thickness of the tool coating does not affect the effectiveness of the boric acid lubricant [9].

Zareh-Desaril et al. [10] showed that nanoparticle additives can improve the tribological properties of lubricating oils. They investigated the tribological behavior of $Al_2O_3$ nanoparticles dispersed in deep-drawn oil during deep-drawing processes. The comparison was made by measuring the cups drawn under different lubrication conditions, observing the surface quality and maximum deformation load. An inverse finite element boundary analysis helped determine the friction coefficient at the tool interface. It was concluded that adding nanoparticles increased the base oil's friction-reducing ability, which reduced the friction between the tool and the test pieces. Another positive conclusion that could be drawn is that the surface quality of the cups formed during the forming process had improved [10].

In addition to plant-based and environmentally friendly alternatives, mineral-oil-based lubricants should also be explored. In addition to the lubricants, the testing technology is also essential, as a well-chosen technology can detect differences between lubricants. A large number of technical papers deal with the Erichsen and cupping tests. Giuliano et al. [11], Kumar et al. [12], and Ramadass et al. [13] also uses the Erichsen test to qualify different lubricants, among many other researchers.

A comparison of mineral-based oils with vegetable oils, which have also been researched by Nurul et al. [14], is of interest because of their excellent properties and low prices, being also popular in industrial applications. In Nurul et al. [14], palm and mineral oils were compared in cold extrusion processes. Their study demonstrated that the palm-kernel oil lubricated test specimens withstood higher extrusion loads than the mineral-oil-based lubricant and did not exhibit significant wear or thinning of the product surface. Their study suggests that mineral-oil-based lubricants used in industry can be replaced by renewable palm oil [14].

In addition to vegetable-based oils, mineral-based oils and dry lubricants are even more widespread in industrial practice, and there is much research on their effects. For example, Demazel et al. [15] found that the quality of deep-drawn parts depends on many parameters such as tool and blank geometry, surface condition, clamping force, forming force, and tooling speed and lubrication. In their research, the swift cup pull (drawing) test [16,17] was used to evaluate the performance of three lubricants. The lubrication performance of three lubricants (two oil-based fluid, and a dry film) applied to AA5182-O discs was investigated. Furthermore, finite element simulations estimated the friction coefficients of these lubricants. The cupping test was found suitable for investigating and modeling the lubrication processes, and the Constellium dry film lubricant could achieve the best friction conditions [15].

Meiler et al. [18] also experimented with mineral-based oils and dry-film lubricants. AlMg4.5Mn0.4 sheet material was formed in an Erichsen testing machine for their tests. For the experiments, they used ALG17- and KTLN16-type mineral base oils explicitly developed for deep drawing and E1-type dry-film lubricant developed for BMW. The GOM ARAMIS system was used for measurements and evaluation of results. During the evaluation, the evolution of the main and secondary deformations and the localization of the maxima on the surface were investigated. The best result was obtained with KTLN16 oil [18].

A large body of research confirms that the Erichsen test is appropriate for classifying lubricants and was used in this study. However, the research also shows that there are still questions to be answered in the research of lubricants applicable to DIC technology, and it will be worthwhile to investigate them further in the future.

The DIC method can be applied in mechanical, civil, electric, and other engineering research, projects, and studies. Among others, they can be related to engineering structures [19–21], railroads [22–29], road construction [30–33], aircraft [34,35], navigation (i.e., shipping) [36], astronautics [37,38], mechanical an automotive engineering [39,40], and so on. In this research, we applied an approach regarding the body sheets of railway and automotive vehicles, and similar research can be found in [41,42]. The relevance of the study being published in this journal is its close connection to transportation [43–45], mechanical sciences [40,46,47], and automotive engineering [39,47–49].

In this paper, Section 2 describes the materials used, test methods, and procedures. Then, the results and conclusions are presented in Sections 3 and 4, respectively.

## 2. Materials and Methods

The main aim of the research was to find lubricants for the sheet metal used in the automotive, railway carriage, and electronics industries that can provide the right friction conditions for formability testing. An important criterion was the selection of materials that are also applied in technological practice and are therefore readily available. In addition to easy availability, the selection of lubricants was also based on the criterion of ease of use and environmental impact. During the tests, thirteen lubricants of different viscosities were used on 0.22 and 0.4 mm thick Al99.5 aluminum thin plates. The material of the sample plate was chosen because it is a material that is commonly used in the automotive industry for various electronic components. Furthermore, thin plates are more sensitive to frictional conditions due to the size effect (grain size vs. plate thickness). In addition to lubricants, solid friction-reducing materials were also investigated, which have the

significant advantages of reusability, no sample contamination, and reduced environmental impact (viscosity parameters were not available for lubricants).

### 2.1. Lubricants and Raw Materials Used in the Research

This section describes the liquid and solid lubricants used in this research.

#### 2.1.1. Lubricants and Oils

The following lubricants and oils were used in this research (Table 1).

**Table 1.** List of lubricants and oils used in this research.

| ID. | Name | Manufacturer (Brand) | Additives |
|---|---|---|---|
| L1 | Polimet EDM 3 | MOL | encrypted by manufacturer |
| L2 | Fortilmo EV 603 | MOL | encrypted by manufacturer |
| L3 | Fortilmo ADD 20 | MOL | encrypted by manufacturer |
| L4 | PTFE DRY | MOTIP | encrypted by manufacturer |
| L5 | PTFE WHITE GREASE | MOTIP | encrypted by manufacturer |
| L6 | UNIVERSAL CONTACT SPRAY | WD-40 | encrypted by manufacturer |
| L7 | Fortilmo EV 671 | MOL | encrypted by manufacturer |
| L8 | Polimet M4 | MOL | encrypted by manufacturer |
| L9 | Fortilmo EV 601 | MOL | encrypted by manufacturer |
| L10 | OXOL-350 SILICONE OIL | T-SILOX | encrypted by manufacturer |
| L11 | GRAPHIT GREASE | PRESSOL | encrypted by manufacturer |
| L12 | HEAVY-DUTY GREASE | LINDE | encrypted by manufacturer |
| L13 | MY E 603 | MOLYDAL | encrypted by manufacturer |

Lubricants were selected from the university's partners based on the general criteria described earlier. The aim was to ensure a uniform biaxial elongation during the test so that the point of rupture would be located in the center of the specimen. The lubricants were applied to each specimen with a brush, in a uniform layer in the center of the specimen in a 40 mm diameter band. Before application, the test specimen surfaces and forming tools were cleaned with isopropyl alcohol to prevent the mixing of the lubricants.

#### 2.1.2. Teflon Foils

Among the solid lubricants, Teflon films are often used in industrial applications, so Kolofol Teflon films, with two different thicknesses (0.22 mm and 0.08 mm), were used in this study. The foils were supplied in rolls and had to be prepared by cutting them into discs for the test. The diameter of the foils was determined according to the dimensions of the working space (inner diameter of the ring was 50 mm, and diameter of the punch was 20 mm). Two sizes were made, 38 mm and 47 mm. The two different sizes were needed to investigate whether the foil's size affected the friction conditions. In determining the smaller diameter, the aim was to cover the 20 mm diameter hemispherical head punch with the foil; thus, a 38 mm diameter was required. The 47 mm diameter disc matched the internal size of the ring but was smaller than it to avoid the possibility of it being squeezed under the binder. The standard for the Erichsen and cupping test (ISO 11531:2022—Metallic materials—Sheet and strip—Earing test, and ISO 20482:2013—Metallic materials—Sheet and strip—Erichsen cupping test) [50,51] refers to the use of foils but does not specify any size or material type restrictions.

During the measurement, the discs were placed between the plate and the punch, so the foil fell in the middle of the specimen. Various layering schemes were also investigated using different oils and greases. This is discussed in detail in Section 3.

#### 2.1.3. Thin Sheets

High-purity and semihard Al99.5 (0.22 mm and 0.4 mm thick) sheet material H14/H24 (EN AW-1050A) was used for this study. For the tests, 90 mm × 90 mm specimens were cut from the sheet coil using a hand-held sheet shear after predrawing, in accordance

with the Erichsen standard (ISO 20482:2013 Metallic materials—Sheet and strip—Erichsen cupping test) [50]. Table 2 contains the chemical composition of the considered, tested aluminum sheets.

**Table 2.** Chemical composition of the test specimens expressed as a percentage by weight.

| Si | Fe | Cu | Mn | Cr | Zn | Ti | Other |
|---|---|---|---|---|---|---|---|
| 0.25% | 0.4% | 0.05% | 0.05% | - | 0.07% | 0.05% | 0.03% |

*2.2. Tools, Equipment, and Measurement Methods Used in this Research*

This section describes the test methods used in this study, the relevant standards, and the measuring instruments and systems. The measurement processes are also described in detail.

2.2.1. Erichsen Test with DIC Evaluation

During the tests, the deflections of the 90 mm × 90 mm Al99.5 specimens with a plate thickness of 0.22 mm and 0.4 mm were measured using a university-developed and -produced hydraulic Erichsen plate testing machine; the tests were performed in accordance with ISO 20482:2013 [50]. The test setup included a 20 mm diameter punch with an associated fixed binder; the binder surfaces had a roughened finish to achieve the maximum possible binder force. The test is continued until the first crack appeared on the surface of the test specimen, while the displacement of the plate displacement is prevented by the fixed binder, so the deformation is subjected to biaxial loading throughout. After the crack's appearance, the punch's relative displacement is recorded in millimeters, with no unit of measurement, giving the Erichsen indentation (*IE*) number. The *IE* number is used to classify the lubricants: the higher this value, the greater the plate deformation; thus, the higher the *IE* number, the better the lubricant's performance. In other words, it can be assumed that the best friction conditions occur in this case. The other important evaluation criterion is the localization of the crack. For each specimen, it was documented whether the crack appeared in the center or off center on the side of the specimen. During the preparation phase, the specimens were primed with white Motip (MOTIP, The Netherlands) heat-resistant acetone-based solvent white paint after cleaning with isopropanol. Then, after a 3 h process of drying, they were coated with a random speckle pattern using Duplicolor matte water-based black paint. The deformation process was recorded using the GOM ARAMIS 5M DIC system, which was positioned above the Erichsen device at a measuring distance of 830 mm.

Before the measurement, the next lubricant was applied or, in the case of foil, placed between the specimen and the punch using the appropriate layering technique. In the end, the lubrication method that provided an above-average *IE* number during the test and started the crack in the center of the specimen was rated good (adequate).

2.2.2. Measurement of the Plate Thickness Run-Off

In the test, the thinning of the deformed plates, i.e., the thickness run-out, was measured. A Mitutoyo ID-S1012XB with a resolution of 0.01 mm and a measuring range of 12.7–0.01 mm, serial number 16037838, was used for the measurements. Before the tests were started, the accuracy of the dial gauge used was checked employing calipers: the instrument had 0.01 mm accuracy. Then, the thickness of the specimens was measured along the axis of symmetry perpendicular to the crack. Finally, the thickness values were read in 2 mm intervals.

During the evaluation of the tests, the results measured by the meter were compared with the results of the DIC measurement. By evaluating the difference, the validation of the DIC measurements could be verified, and the correct qualification of the lubricants could be checked. However, when evaluating and checking the results, the thickness of the paint layer was considered, and the measurement results were corrected for this thickness.

### 2.2.3. GOM ARAMIS

During the tests, the deformations on the surface of the plates were recorded using a GOM ARAMIS 5M DIC measuring system. GOM ARAMIS technology, based on the digital image correlation principle, generates a series of images of dynamic stresses by measuring the specimens' 3D coordinates, displacement, and surface deformation. Owing to advances in camera technology, high-resolution 12-megapixel photos can be taken [52,53]. The measured material can be also metals, metal matrices, composites, etc. [53].

Owing to the camera's high resolution, the system can capture microscopic changes over a large surface area, providing accurate measurement conditions and optimal camera calibration are applied. The system has adjustable and interchangeable lenses, LED illumination, and polarization filters. The instrument can be adjusted to the measurement conditions owing to the many adjustable accessories, and positioning is quick and easy owing to mobile accessories. Because the measuring conditions—temperature, light conditions, and humidity—are not always the same, the equipment requires calibration at fixed intervals (usually every 8 h). In addition, it is necessary to ensure the correct measurement accuracy. These considerations apply to any adjustable camera system designed for a development or research environment. However, the fixed camera image correlation systems designed for industrial use are more stable and require less sensor calibration. At full resolution, the maximum image capture rate can be up to 1000 frames per second (fps) [46].

The purpose of the measuring system is to record the deformation of the load as a function of time using random patterns painted on the surface of the plate. Therefore, the quality of the pattern is of paramount importance for the outcome of the measurement. A good-quality pattern reduces image noise and increases contrast and measurement resolution, thereby increasing the accuracy of the measurement results. The correct density of 'dots' in the pattern is also an important criterion, as it determines the degree of contrast and blur.

During the imaging mechanism, the system captures the deformation through a small image window and an evaluation window (facet and subframe, respectively). The user can monitor the imaging process of the measurement system, which compares the evaluation window with the reference images, while the operator makes sure that the evaluation window does not move at any point during the measurement. The degree of fit is quantified from the similarities between images and the changes in the gray level. The evaluation window is responsible for detecting the relationships between the images in the image sequence, which it identifies due to the unique pattern. DIC software uses the Levenberg–Marquardt algorithm [54–57] to nonlinearly optimize the deformation of the evaluation window for adequate accuracy. The window deformation is due to changes in measurement conditions caused by external influences. The functional methods, including the ARAMIS system's calculation process, can be found in the machine's user manual.

The GOM ARAMIS system used in this study was an ARAMIS 5M system with 5-megapixel cameras. The cameras were mounted on an 800 mm long adjustment slide, equipped with 2 LED light sources. The measurement frequency could reach a maximum of 25 Hz, but a sampling rate of 9 Hz was sufficient in the tests. A CP20 measuring range of 120 mm × 100 mm with a focus of 60 mm was used for the measurement, which is sufficient for the deformations achievable with the Erichsen test. After the measurements, the results were evaluated in ARAMIS 2018 software.

## 3. Results and Discussion

This section presents the results of the tests described in the measurement procedures. For each test, the lubrication methods that performed well and poorly in that test are highlighted.

### 3.1. Results of the Erichsen Test

The cut-to-size 90 mm × 90 mm specimens were cleaned and marked on the symmetry axes (x, y) every 2 mm, which will be used for the thickness run. After marking, the Erichsen

tests were performed with the 20 mm forming punch with fixed crease inhibition, where the forming was continued until the first crack appeared. At the end of the measurement, the relative punch displacement was recorded in the measurement database, followed by the derived *IE* number. The results sorted and evaluated by *IE* numbers are shown in Table 3.

**Table 3.** Erichsen test results: *IE* numbers based on cumulative averages of 0.22 mm and 0.4 mm plates.

| ID. | Name | Manufacturer (Brand) | *IE* Number |
|-----|------|----------------------|-------------|
| L12 | HEAVY-DUTY GREASE | LINDE | 27.58 |
| L11 | GRAPHIT GREASE | PRESSOL | 27.30 |
| L5 | PTFE WHITE GREASE | MOTIP | 27.28 |
| L10 | OXOL-350 SILICONE OIL | T-SILOX | 27.20 |
| L3 | Fortilmo ADD 20 | MOL | 27.10 |
| L8 | Polimet M4 | MOL | 26.83 |
| L9 | Fortilmo EV 601 | MOL | 26.80 |
| L13 | MY E 603 | MOLYDAL | 26.68 |
| L2 | Fortilmo EV 603 | MOL | 26.60 |
| L7 | Fortilmo EV 671 | MOL | 26.60 |
| L6 | UNIVERSAL CONTACT SPRAY | WD-40 | 26.38 |
| L1 | Polimet EDM 3 | MOL | 26.30 |
| L4 | PTFE DRY | MOTIP | 26.28 |

The results presented in Table 4 show that the best-performing sample was lubricant 12 (i.e., L12), which achieved the highest *IE* number and, thus, the most extensive deformation. It can also be seen that the top three places are occupied by grease; i.e., greases uniformly performed better than oils or even dry Teflon spray, which performed the worst. As a reference, an Erichsen test was also performed without lubrication, with an average *IE* number of 24.11. It can be seen that the worst lubricant (L4) increased the *IE* number by 9%, while the best (L12) increased it by 14.39%. The results show that lubricating greases created better friction conditions than oils. The oil layer was probably more quickly displaced from the contact surfaces of the punch and the workpiece due to surface pressure, while the grease was better able to remain there. In formability testing, friction must be kept to a minimum, so the results suggest that greases are preferable. However, the rupture picture is not yet favorable, and these results are presented in Section 3.2.

**Table 4.** Results of the Erichsen test with Teflon films (based on averages of 0.22 mm and 0.4 mm plates).

| ID. | Name | Brand | *IE* Number—0.08 mm Teflon | *IE* Number—0.22 mm Teflon |
|-----|------|-------|----------------------------|----------------------------|
| L1 | Polimet EDM 3 | MOL | 27.60 | 26.80 |
| L2 | Polimet EV 603 | MOL | 27.40 | 26.90 |
| L3 | Fortilmo ADD 20 | MOL | 27.90 | 27.00 |
| L4 | PTFE DRY | MOTIP | 27.20 | 27.00 |
| L5 | PTFE WHITE GREASE | MOTIP | 27.40 | 27.20 |
| L6 | UNIVERSAL CONTACT SPRAY | WD-40 | 26.40 | 27.10 |
| L7 | Fortilmo EV 671 | MOL | 27.90 | 27.10 |
| L8 | M4 | MOL | 27.90 | 27.40 |
| L9 | Fortilmo EV 601 | MOL | 27.60 | 27.00 |
| L10 | OXOL-350 SILICONE OIL | T-SILOX | 27.60 | 28.00 |
| L11 | GRAPHIT GREASE | PRESSOL | 28.50 | 28.00 |
| L12 | HEAVY-DUTY GREASE | LINDE | 29.00 | 28.30 |
| L13 | MY E 603 | MOLYDAL | 27.20 | 26.80 |

The Erichsen tests were also carried out with Teflon film discs of 0.08 mm and 0.22 mm thickness, where the lubricants were used together with the Teflon films. The application

of the lubricants was evenly distributed in the center of the test specimen according to the size of the disc. The results of the tests are shown in Table 4.

The results show that using teflon films for the best lubricants (L12, L11, and L5) increased by an average additional 5%. However, the experiments showed that the 38 mm discs slipped, while the 47 mm discs remained centered. The slippage caused several measurement failures due to curling, so experiments were only performed with the 47 mm discs. In summary, lubricants L12, L11, and L5 could achieve better *IE* figures, i.e., better lubricity.

### 3.2. Recording of Thickness Run-Off Curves from Specimens Made with the Erichsen Test

After evaluating the *IE* numbers of the Erichsen tests, the thickness run-off curves were determined. The measuring points, marked with 2 mm spacings previously applied to the specimens, followed the deformation of the specimen during the tensile test. Plate thickness values read at the markings were measured with a dial gauge according to the procedure described in Section 2.2.1. The symmetry line drawn in the rolling direction was marked "y", and the symmetry axis perpendicular to it was marked "x".

Figure 1 shows one of the specimens after cracking with the drawn symmetry axes and the measurement points.

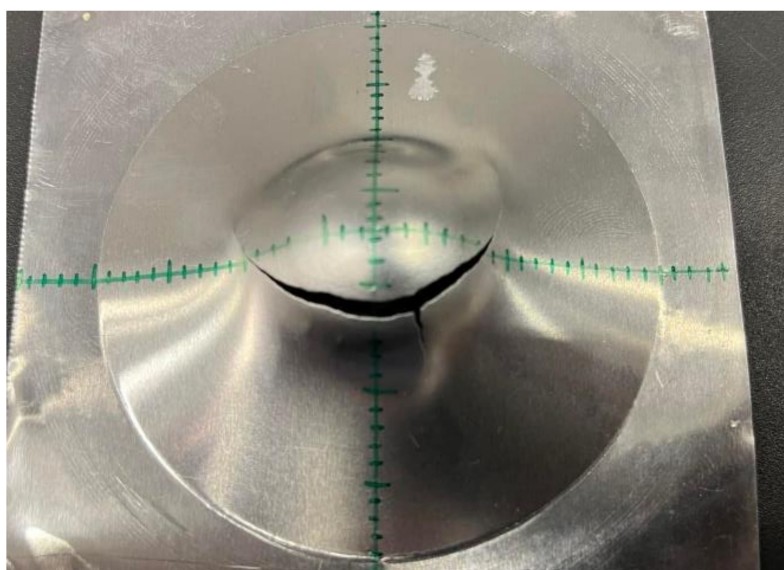

**Figure 1.** Specimen for thickness run-off test.

Figure 2 shows a sample evaluation with one of the poor-performing L4 lubricants and the two Teflon foils.

Figure 2 clearly shows that when using the lubricant only, the crack was not centered, which can be seen from the minimum thickness being not centered; two minimum values were offset from the center. It occurs because the lubricant is displaced between the punch and the plate, and at the end of the forming process, the plate will flatten at the top of the punch, and a ring-shaped thinning will occur, and then the crack will appear there. It can also be seen that both teflon films have improved the original condition.

Figure 3 shows the results for sample L12, similar to the previous one, using only the lubricant and the 0.08 mm and 0.22 mm Teflon films. It can be clearly seen that a higher thinning could be achieved by using the L12 lubricant, and Teflon discs could even improve on this value. It can also be observed from the diagram that the thickness minima were not in the middle here either and that there was also a ring-shaped thinning so that the disc flattened at the end of the forming process. The better thinning and higher *IE* number occurred because the lubricating grease could stay longer between the surfaces, so the flattening occurred later.

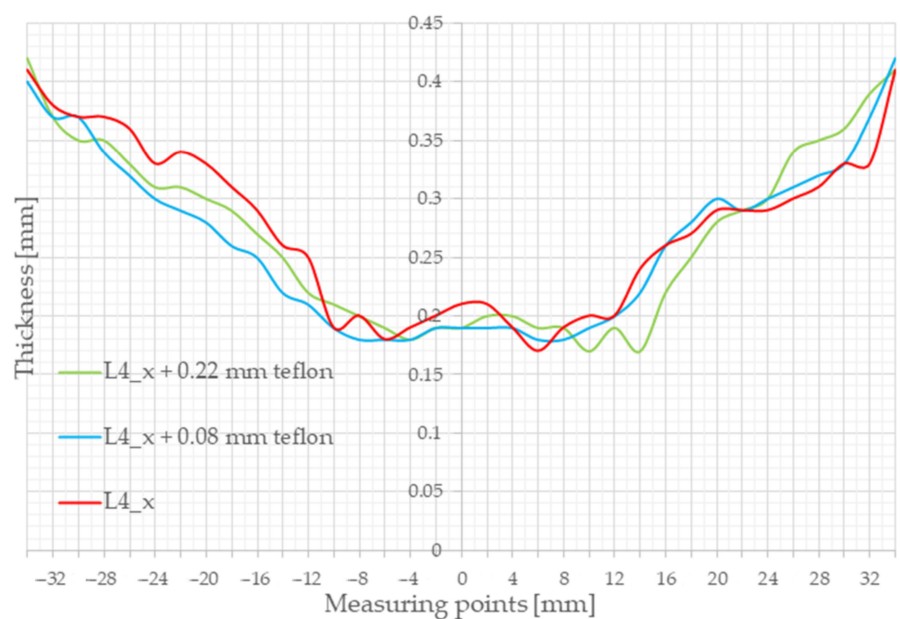

**Figure 2.** Thickness run-off with sample L4 on 0.4 mm plate, measured with a dial gauge.

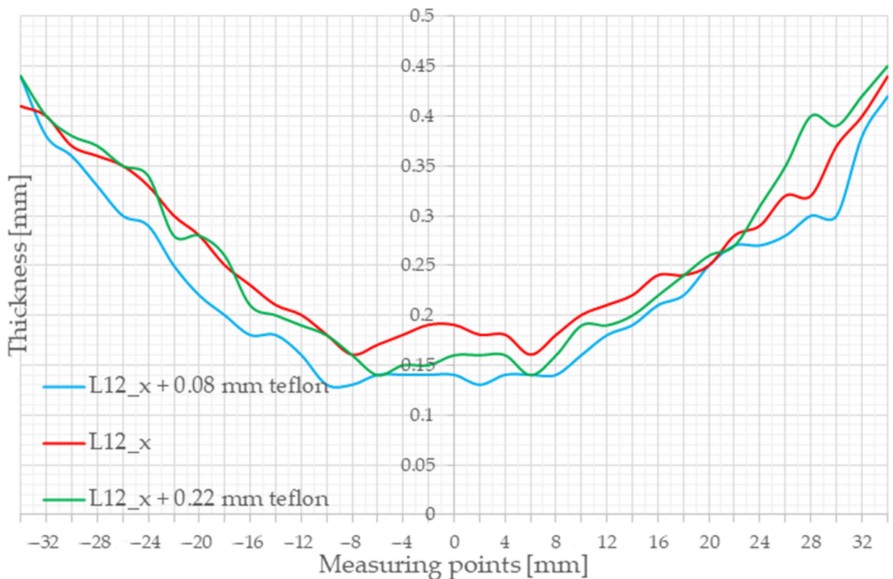

**Figure 3.** Thickness run-off with sample L12 on 0.4 mm plate, measured with a dial gauge.

Based on the evaluation of the thickness curves on all samples, we concluded that there was a correlation between the *IE* number and the thinning because the ranking of the lubricants from the curves was similarly formed when evaluating the *IE* numbers. This result was expected, as similar conclusions have been reached in previous studies. The top three best-performing lubricants were L12, L11, and L5; the worst-performing lubricant was L4. We also found that better elongation values could be achieved using the thinner Teflon film (0.08 mm).

For each specimen, it could be observed that at the end of the forming process, the thinning was not at the center of the specimen. In other words, the crack did not appear in the center, which is essential when recording a standard forming-limit diagram. Using Teflon films will significantly improve formability but will not eliminate the end-of-forming chipping.

*3.3. Results of the DIC Test*

DIC technology was applied in the following research phase to investigate the localization of thickness reductions and to study friction processes during forming. Using a GOM ARAMIS system, real-time images could be acquired, the entire forming process only be studied, the crack propagation process only be visualized, and the stress images only be used to infer the friction conditions.

After cleaning with isopropanol, 90 mm × 90 mm specimens were primed with Motip heat-resistant white matte paint, and after 3 h of drying time, they were spot-patterned with Doplicolor matte black paint according to the CP20 measurement range. After patterning, the specimens could be clamped and shaped.

It was essential to check and validate the system before starting the DIC tests. A 0.22 mm thick specimen plate was loaded to crack in the Erichsen machine, while the deformation was recorded with an ARAMIS system. After the measurement, thickness values were extracted from the ARAMIS system along the axis of symmetry perpendicular to the crack, and the thickness deflection curve was recorded on the same test. The results were then compared. Figure 4 shows the validation results. The curves are almost parallel, with differences within a margin of error of 0.02 mm, which is the same as the error of the dial gauge. Based on the results, the ARAMIS system can be considered validated.

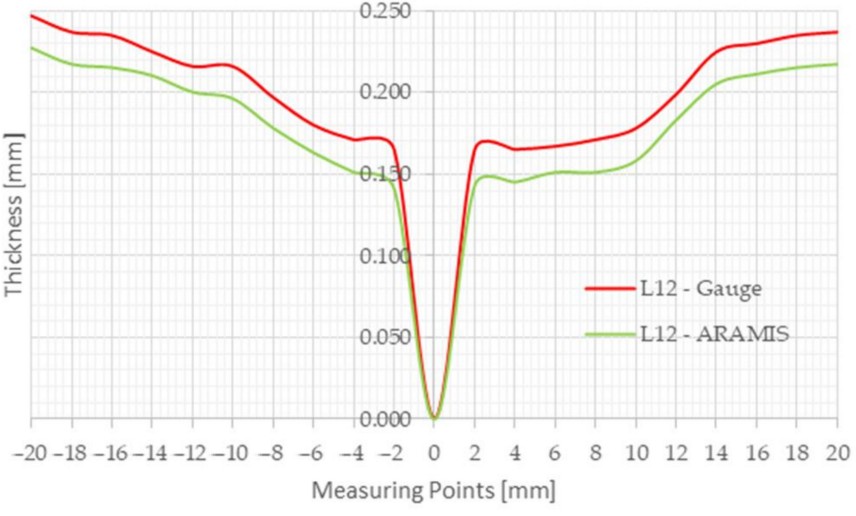

**Figure 4.** Validating the results of the ARAMIS system.

After validation, the recorded images were checked. The test's purpose was to visualize the details of the forming history. After evaluation of the measurement results, the following typical stress image was obtained. Figure 5 shows the result of the ARAMIS measurement on a 0.22 mm plate with L5 lubricant. The figure clearly shows the thinning in the form of a red ring. The color plot scale represents the thickness reduction in the plates from red (thin) to blue (thick). This pattern was present for all lubricant combinations to varying degrees. The DIC test was able to confirm that the lubricants were displaced, to varying degrees, between the plate and punch at the end of the forming process so that the crack did not appear in the center, i.e., the maximum thinning did not occur at the origin.

A further task in this study was to optimize the crack location. Because the results showed that even the best three lubricants with Teflon film could not push the crack to the center, the next step was to investigate the layer order of the lubricants. This was worth trying because using the Teflon films allowed the L12, L11, and L5 lubricants to crack near the center. As previous tests showed that 0.08 mm Teflon film gave 1–2% better results than the 0.22 mm film, the following lubricant layering was tested: lubricant + 1 piece 0.08 mm Teflon film + lubricant + 1 piece 0.08 mm Teflon film + lubricant. This layering arrangement only caused a minimal increase in diameter on the punch surface, so measurements could still be made in the standard way. One of the results of the layering test is shown in Figure 6.

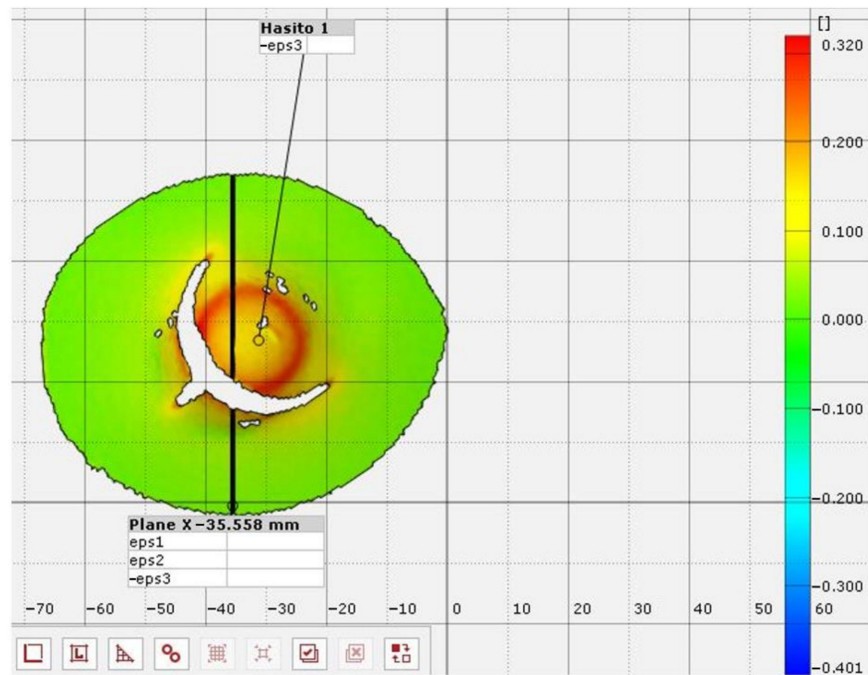

**Figure 5.** Formation of the ring-shaped stress field in the case of lubricant L5. The legend on the right shows values in millimeters.

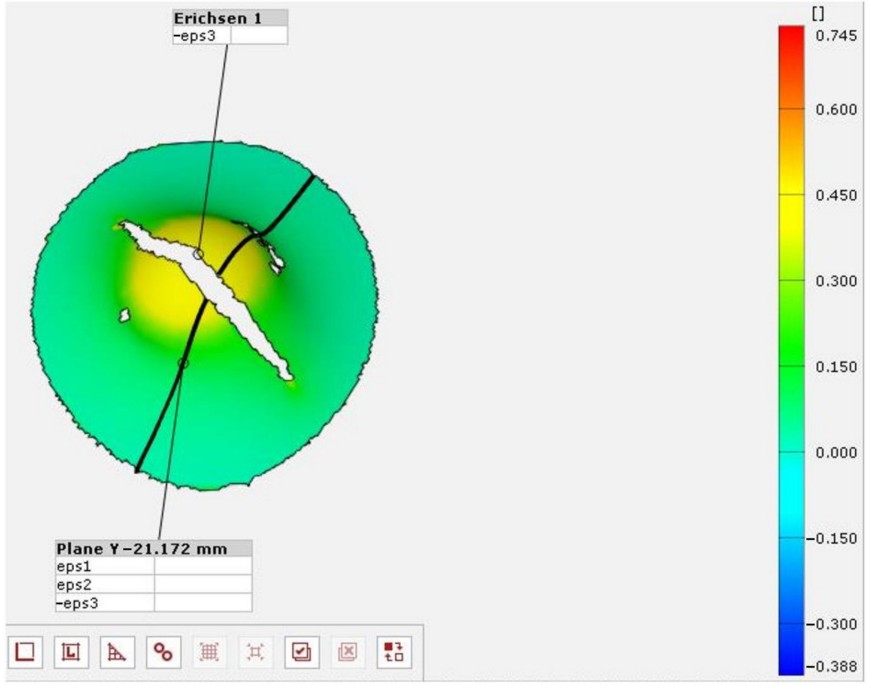

**Figure 6.** Image of the specimen torn in the middle, in the case of lubricant L12, measured with an ARAMIS system. The legend on the right shows values in millimeters.

The test results were successful because all three greases (L12, L11, and L5) in the given layer order caused the specimens to tear in the middle, the thickness minima to appear at the origin, and annular stress to not appear. However, for the oils, even with the modified lubrication, the cracks could not move to the center, and the results improved (middle image in Figure 7), but the annular stress, i.e., the end of deformation chipping, did not completely disappear.

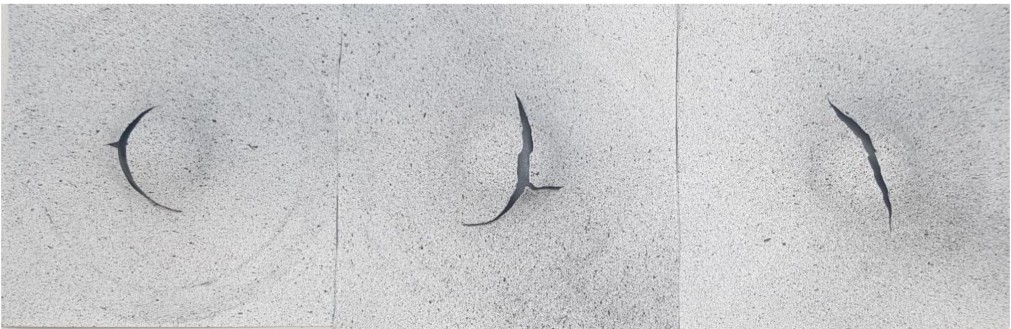

**Figure 7.** The result of the optimization process.

Figure 7 shows the result of the optimization of the lubrication process. It can be seen that, in contrast with the oils (left and center figures), the lubricating greases in the applied layering scheme could push the crack to the center of the specimen. Based on these results, making a standard forming-limit diagram (FLC) recording on thin aluminum plates is safe because the friction coefficients no longer significantly influence the measurement results, reducing the measurement error because the crack will appear in the middle of the specimen. In this case, it can be assumed that the biaxial deformation will be uniform.

## 4. Conclusions

The results of the study showed that for the lubricants and oils tested, there was a 5% difference in *IE* diagrams between the best (L12) and worst (L4) cases, which is a significant difference for thin plates.

We found that this value could be increased and improved by using Teflon discs. In addition, the Teflon discs did not squeeze out of the surface of the punch and the tested plate due to the forming pressure, so they could maintain the lubrication throughout, thus further increasing the elongation values and the formability of the plates while reducing the degree of hardening.

The localization of stress maxima, i.e., the centering of cracks in the specimen, could be achieved by combining Teflon discs and L12 lubricant (with the appropriate layer order), which significantly aided in the recording of standard FLC diagrams. For a successful standard measurement, it is essential that the crack is located in the center of the specimen, so the results of this study can be used for further research and technological investigations. Using foils is also advantageous because they are readily available, have no expiry date, and are of less environmental concern.

**Author Contributions:** Conceptualization, S.S. and H.C.; methodology, S.S.; software, S.S. and H.C.; validation, S.S. and H.C.; formal analysis, S.S. and H.C.; investigation, S.S., H.C., D.K., A.N., M.S. and S.F.; resources, S.S., A.N. and S.F.; data curation, S.S. and H.C.; writing—original draft preparation, S.S., H.C., D.K., A.N., M.S. and S.F.; writing—review and editing, S.S., H.C., D.K., A.N., M.S., and S.F.; visualization, S.S. and H.C.; supervision, D.K., A.N., M.S. and S.F.; project administration, S.S.; funding acquisition, S.F. All authors have read and agreed to the published version of the manuscript.

**Funding:** This research received no external funding.

**Data Availability Statement:** Not applicable.

**Acknowledgments:** This paper was prepared by the SZE-RAIL research team.

**Conflicts of Interest:** The authors declare no conflict of interest.

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
