# Peer review of "Testing of Lubricants for DIC Tests to Measure the Forming Limit Diagrams of Aluminum Thin Sheet Materials"

_infrastructures, doi:10.3390/infrastructures8020032_

Round 1

Reviewer 1 Report

In this work, the authors have presented an experimental comparison among some lubricants which are commonly used in metal forming. 

They combine DIC 3D and dials to measure the displacements and thickness distribution after the test. They conclude with some improvements in the process when specific lubricants are used.

The introduction have been properly performed. The methodology and the conclusions are clear.

The paper is well-organized and the results supports the discussion.

The manuscript deserves attention of the community and can be considered for a publication after minor corrections listed below.

General comments:

It is recommended to write the standards title in the references. Review the Line 173 and Line 185, and try to write them as Line 96.

The point number 2.2.1. is repeated. In line 220, it should be written 2.2.2. and check the cross-reference in line 327 to this point.

The method to measure the thickness distribution using DIC 3D is nos clearly described. DIC 3D reports the displacements field of the painted surface. But, how do you calculate the difference in position (i.e. out of plane position) between each point and the equivalent point at the opposite face? Do you assume that the punch is infinitely stiff? Then, is the any implementation to perform this calculation?

The Erichsen test equipment has not been referenced. Could you specify the brand and model of this equipment?

Author Response

See the attached PDF file. Please check the other PDF file in which all the modifications have been tracked.

Reviewer 2 Report

General Comments

The authors present the article “Testing of lubricants for DIC tests to measure the forming limit diagrams of aluminum thin sheet materials.” The article presented is exciting and promising. However, the results presented in the paper are of some value, but their presentation effectively disrupts their reception.

Major comments

1. The authors should expand why the selection of these 13 lubricants that were tested as it is the central part of the publication.

2. The figures used throughout the publication must be restarted.

Figures 1, 2, 4, and 5 have no further contribution to the publication.

Figures 7, 8, and 9 are from the ARAMIS platform but do not have the same format as the publication. These must be in the format of the publication. In this way, have the same quality as the publication.

Figures 10 and 11, which are from the same platform, should explain the entire context of the image. Nevertheless, unfortunately, not all potential readers have access to ARAMIS, which makes them unable to understand the meaning of what is in the image.

Minor comments

3. In the last paragraph of the introduction, it must add how the article will be structured.

4. Authors must add a methodology section. This section will give future readers, including graduate students to follow in the footsteps of the presented research.

5. In section 2. Materials and methods in lines 142-143, viscosities are discussed. Readers must know what ranges of viscosities are being handled with the selected lubricants and what influences their use will have.

6. In section 2.1.2 Teflon foils in line 164, is the thickness 0.22 mm or between 0.22 mm at 0.08 mm?

Final Comments

The idea presented and developed in the paper seems to be promising and interesting, but due to the above concerns, at this moment I do not recommend the paper for publication.

Author Response

(The authors gave the same response as above.)

Reviewer 3 Report

The work explore different commerical lubricating solutions to measure forming limit of Al sheet with DIC technology.

the authors ; 

However, fails to presents chemicals additives information of lubricants employed and how their effective comparsion of lubrication performance. 

Authors can see some of suggestions mentioned below:

 1. Please rewrite the abstract to provide more comprehsive Subjective information. 

 2. Authors can improve the introduction by increasing some of recent works on metal forming using different nanolubricants. 

3. It would be great if The crux of Yoshimara studied can summarsied in 2-3 lines rather 2 paragraph "line 51-62"

4. It advised to remove Table 1 showing "Possible uses of different vegetable oils"

Since one can misunderstood its potential application. and does not provide any addition to studies. 

5. What doe sauthor inteds to say here "In this research, the authors applied an ap-134 proach regarding railway and automotive vehicles and their car body sheets"

Please simplify?

6. Please include the selection of taken lubricants in perspective to its correlation with indeed material for your study ? 

Please modify table 2. with potential additives in oils selected?

8. If WD-40 selected is used as forming lubricants ? 

Figure 5 is unncessary can be removed

Authors can include the schematic of Erichsen test which could be a better representative for understanding in fig 4.

It would be great if authors can summarized the section 2.2.3. GOM ARAMIS shorter with key details only.

Please do the mention about IE numbers? lower or higher which is desired ?

Authors should discuss more about better performance of (L12, L11, and L5) lubricant; might be perhaps to better tackiness  or high viscosity, or load bearing.

Authors can refer to https://doi.org/10.1016/j.triboint.2020.106302" in regards to PTFE

Please elaborate more on friction "Based on these 419 results, making a standard FLC (forming limit diagram) recording on thin aluminum 420 plates is safe since the friction coefficients no longer significantly influence the measure-421 ment results, reducing the measurement error."why authors think such ?

Kindly elaborate on your conclusions section.

Author Response

(The authors gave the same response as above.)

Round 2

Reviewer 2 Report

Dear Authors 

After reviewing the changes made in the manuscript and the response to each point by the authors, I finally recommend publishing this article. 

Kind regards,

Reviewer

Reviewer 3 Report

The authors have made essential changes suggested previously and improved the manuscript. It can be considered for publications